# Simulation Analysis of the Efficiency of Evacuation Entrances of Strip-like Metro Commercial Streets: A Case Study of Wuhan Business District Metro Station

Hong Xu [1,2,*] and Meixi Chen [1]

1   School of Urban Construction, Wuhan University of Science and Technology, Wuhan 430065, China;
    merxy@wust.edu.cn
2   Hubei Provincial Engineering Research Center of Urban Regeneration, Wuhan University of Science and
    Technology, Wuhan 430065, China
*   Correspondence: xuhong@wust.edu.cn

**Abstract:** The high-performance evacuation of commercial streets is the primary requirement for subway planning and construction. This study introduces a simulation and analysis framework to analyze the evacuation efficiency of a strip subway commercial street with the same number of entrances and exits and different entrance positions. The study modifies the entrances/exits of strip-shaped metro business street plans in a simulation approach to analyze the variation in evacuation effectiveness under varying exit layouts, as well as the effects of different escape entries on the subway commercial street. The study aims to find the optimized spatial structure of evacuation entrances of strip-like metro commercial streets. In this simulation approach, the scenarios of different combinations of entrances and exits, the floating scenario of the number of people to be evacuated, and the scenario of adjusting the proportion of elevator personnel were simulated. Through comparing the evacuation times of these simulated layout scenarios, this study finds that equidistant interval insertion-type entrance and exit arrangements of strip-shaped metro business street plans are better suited to improving evacuation efficiency. This finding can be used as a manual to improve the evacuation capacity of subway commercial streets, which will be helpful for the planning and design of subway commercial streets.

**Keywords:** evacuation effectiveness; plan design; Pathfinder; strip-shaped metro business street





## 1. Introduction

Land resources are becoming scarcer due to the acceleration of urbanization and social development. Underground space building will progressively take the lead as an urban development strategy [1] in developing countries like China. For example, China has now developed many underground spaces in terms of the underground, underground complex, and other elements [2]. Meanwhile, the subway has taken over as a primary mode of urban transportation, allowing the quick movement of people, goods, services, and information. In particular, the space linking the subway, the underground, and commercial streets facilitates a three-dimensional business circle effect. Within this kind of space, the flow of people is more complex and multi-directional, and these flows can easily conflict with each other. Therefore, how to construct a reasonable and effective underground commercial street space plan that conforms to the fastest evacuation of people is a crucial issue in the design and research of underground space commercial complexes.

Path selection for evacuations and simulating pedestrian behavior have always been hot problems, and simulating pedestrian behavior has always been a significant method of theoretical research in this area. Macroscopic and microscopic models are typically used to describe crowd evacuation scenarios [3]. The macroscopic model, which focuses on the overall pedestrian speed and density, is based on fluid mechanics. Microscopic

models emphasize the interactions between individuals and treat people as free-floating particles. Common microscopic models include the social force model [4,5], the cellular automata model [6–8], the lattice gas model [9,10], and so on. Swarm intelligence algorithms and their optimization [11–13] and deep learning algorithms [14,15] are the key path-planning techniques. Additionally, there are still many software systems like Pathfinder and AnyLogic that can replicate the path taken by pedestrians during an evacuation. The selection of entrances and exits has consistently been the subject of study. Pedestrians may push each other as they flee and compete for safety exits, leading to violent instances like trampling. Any alteration in the crowd's morphology will result in additional secondary catastrophes [16–19]. The pedestrian evacuation capacity gap is fully considered in [20], which simulated the exit choice based on game theory and took into account that the pedestrian escape decision involves a process of contact with others. In the aspect of disparities among people's spatial cognition of buildings and evacuation effectiveness, Li [21] concluded that people would self-estimate the fastest route and choose to avoid danger based on their physical and mental experiences. According to different exit widths and doors, Li et al. [22] examined a variety of evacuation-related factors and concluded that the symmetrical double exit layout was preferable to the asymmetrical double exit layout and that the evacuation was most favorable when the exit width was 1.1 m. Zhang et al. [23] simulated the interaction between leaders and followers and discovered various guiding tactics and determined the best guidance strategy to increase evacuation effectiveness. The analysis of the evacuation of a room or a fire compartment [24] shows that the symmetric distribution on both sides of the exit position may have increased positively evacuation efficiency. This study will conduct a simulation analysis of the efficiency of evacuation entrances of strip-like metro commercial streets within a more complicated space.

Several studies have examined the evacuation procedure of stations by establishing various safety evaluation index systems [25], and they have examined the co-evolution mechanism of commercial spaces in station areas from the aspects of the degree of agglomeration, the spatial distribution, and the format composition [26]. These studies have also examined the characteristics of the comprehensive development of various forms of urban rail [27,28]. According to Zhang et al. [29], the commercial development of a connected underground area will draw more passengers, which will harm the station's evacuation efficiency. The passengers in the station can be directed with the help of video surveillance and emergency broadcasting. Mu et al. [30] examined the evacuation situation of a subway commercial strip during rush hour using the fine grid computing simulation method. Even though these studies have conducted extensive research on the evacuation of subway space, they have largely been restricted to the safety of subway operation space and have not conducted extensive research on the structure of comprehensive underground space organization combined with commerce. Li et al. [31] analyzed the effect of the number and location of evacuation exits on evacuation capacity using an optimized RVO algorithm, they selected fire compartments in a single underground commercial street as the subject. Their works need to be further extended to be necessarily applicable to the overall metro commercial street.

With the use of subways becoming more widespread and efficient, the future underground entrances and exits will no longer be a single or a few arrangements, but an evacuation situation with many uncertain conditions, the spatial accessibility of the underground commercial street evacuation entrances and entrances and pedestrian route selection will be very different optimal spatial structure of evacuation entrances of strip-like metro commercial streets is difficult to capture without a practical simulation and analysis framework. Therefore, this study introduces a simulation and analysis framework to analyze the evacuation efficiency of a strip subway commercial street with the same number of entrances and exits and different entrance positions. The study modifies entrances/exits of strip-shaped metro business street plans in a simulation approach to analyze the variation in evacuation effectiveness under varying exit layouts as well as the effects of different escape entries on the subway commercial street. This study aims to find the optimized

spatial structure of evacuation entrances of strip-like metro commercial streets. In this study, the complete subway commercial street plan of Wuhan's business district is chosen as the study area, and the typical strip plan of an existing subway commercial street is chosen for simulation scenarios using the evacuation simulation software, Pathfinder. Concerning the arrangement of subway commercial streets in China, this study implements a simulation analysis to find a reasonable and effective underground commercial street space plan that conforms to the fastest evacuation of people, which is important in layout plan guidance compared to the previous architectural simulation.

## 2. Study Area

This study defines a typical metro commercial street plan with shops and supporting facilities on both sides of a single corridor as a strip-like metro commercial street. Based on the realistic existence of many strip-like metro commercial streets in China, the study selects the Wuhan business district metro station as a typical strip-like metro commercial street study area. The Wuhan Business District Station (Figure 1) is situated in Wangjiadun, Jianghan District, Wuhan, China. It serves as a transition hub for Wuhan Rail Transit Lines 3 and 7. Overall, it is separated into four districts, and the north side of its hall has a commercial street. The first-floor subway station and the entire commercial street are both 5 m above ground level. The building is about 278 m in length and 20 m in breadth and has a total construction area of about 6308 m, including the front chamber for evacuation and the sinking square. The subway station has eight entrances and exits on this side of the commercial street.

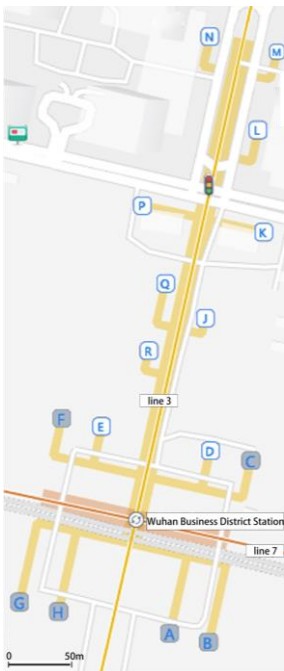

**Figure 1.** Schematic map of each exit of the subway station (source: the Gaode Map).

The subway station serves as a transfer hub station and experiences heavy human traffic. Numerous commercial office buildings are located next to the subway commercial strip on its northern side, and after crossing Huaihai Road, numerous residential structures and the sizable Wangjiadun Park are visible. A daily convoluted stream of people passes through the subterranean commercial street, including workers, locals, and visitors to the park. The subway station's design is "Bright Star River". In the station's lobby, there is a sizable LED lighting dome that resembles fireworks. Therefore, the Wuhan Business District subway station is referred to as "the most beautiful subway station in Wuhan" because it is one of its widely advertised attractions. Some people have been drawn here by the creative

designs and raucous reputations, and the food options in the subway station's commercial street have grown in importance to tourists. Due to this, there is no discernible difference between non-holiday and holiday crowds in the underground commercial strip in Wuhan Business District, and the highest value is evident in the morning and evening. The Wuhan Business District's subway commercial street is generally a sophisticated subway commercial street. A schematic of an evacuation from a subway is shown in Figure 2. Analyzing and simulating this subterranean business street has great research value.

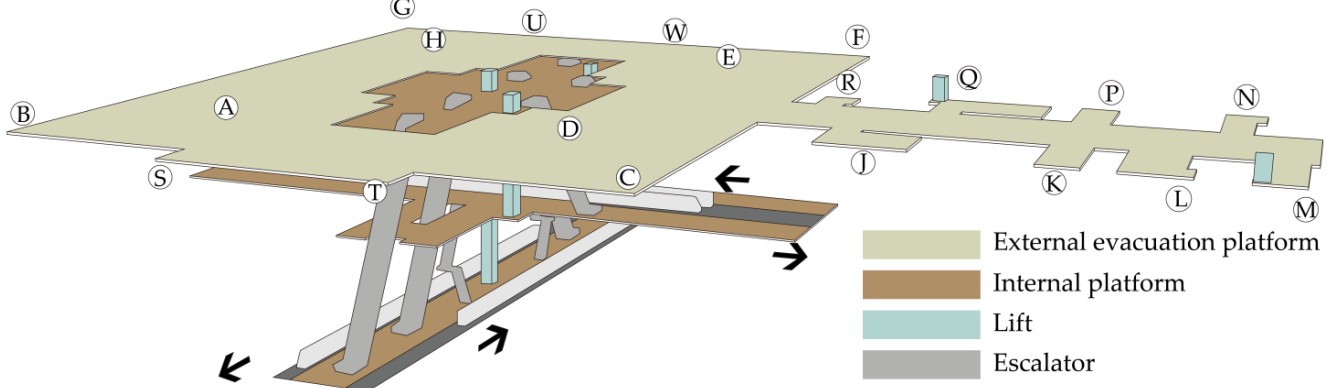

**Figure 2.** Simulation space of subway evacuation.

This study creates a pertinent schematic diagram to gain a more intuitive knowledge of the underground commercial street in the Wuhan Business District. An illustration of the subterranean business street is shown in Figure 3. The entire commercial street is strip-like, with stores on either side, an evacuation area in the middle, and entrances, exits, and sinking squares scattered throughout. Figure 4 shows a schematic representation of fire protection zones segmented by fire shutters and fire doors. Only the stairs included in the single-function part of the subway are not simulated because this article only analyzes the layout planning of the entrances and exits of the commercial street component of the subway. And according to our field research, Table 1 shows the number of stair lifts used for evacuation at each entrance and exit, as well as their width.

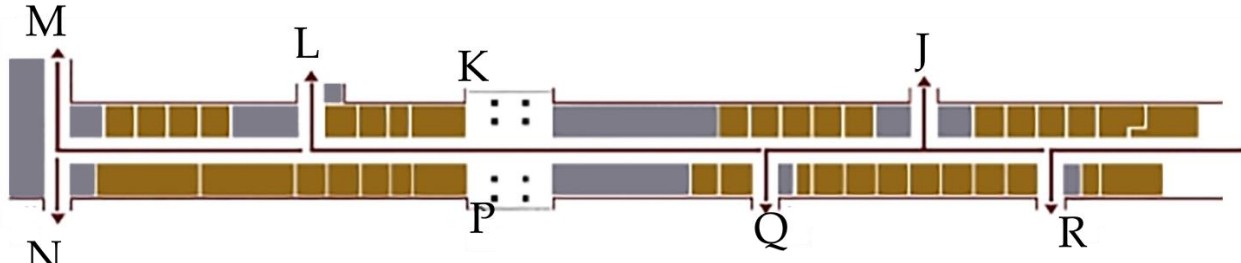

**Figure 3.** Schematic plan of subway commercial street.

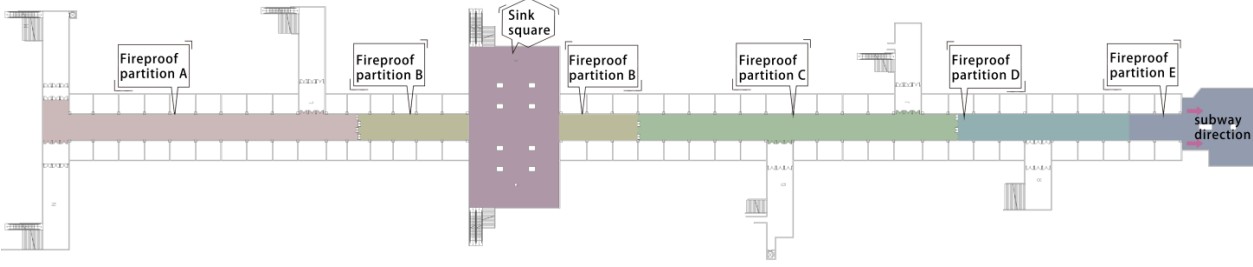

**Figure 4.** Fire prevention zoning of subway commercial street.

**Table 1.** Composition and width of different evacuation openings.

| Entrances and Exits | Evacuation Port Composition | Evacuation Opening Width (m) | Corresponding Region |
|---|---|---|---|
| M | Go up the escalator<br>Staircase<br>Lift | 1.6<br>5.6<br>1.6 | Fireproof partition A |
| N | Descend the escalator<br>Staircase | 1.6<br>4.8 | Fireproof partition A |
| L | Go up the escalator<br>Staircase | 1.6<br>4.8 | Fireproof partition A |
| K | Go up the escalator<br>Descend the escalator<br>Staircase | 1.6<br>1.6<br>3 | Fireproof partition B |
| P | Go up the escalator<br>Descend the escalator<br>Staircase | 1.6<br>1.6<br>3 | Fireproof partition B |
| Q | Staircase<br>Lift | 3.2<br>2 | Fireproof partition C |
| J | Descend the escalator<br>Staircase | 1.6<br>3 | Fireproof partition C |
| R | staircase | 6.4 | Fireproof partition D |

## 3. Simulation and Methods

### 3.1. Methods

The choice of evacuation path is related to the availability and accessibility of entrances and exits and the complexity of the layout [32]. The commercial street plane of the Wuhan business district subway station is the most basic inner-corridor strip plane, which is much less complex. Given that it is the same plane oriented to a simple type, it has little influence on the relevance of evacuation, and therefore, this study focuses on the availability and accessibility of entrances and exits. In addition to the entrance/exit factor of the commercial street, there is also a people factor that affects evacuation efficiency. Studies [23,33] show that crowds can exhibit fleeing behavior, competing behavior, inertial behavior, panic behavior, etc. And the fleeing density of the crowd has a direct impact on evacuation efficiency [3,7,33]: the more crowded the crowd is, the slower the evacuation speed. The composition of the evacuees is also relevant. Through their analysis, the authors of [34] found that the older the age, the longer the pre-evacuation time, and that gender does not have a significant impact on the pre-evacuation time as the behavior and reaction speed of adult males and females in the crowd are relatively rapid. Moreover, the gap for disadvantaged groups including the elderly, children, and disabled people, in the early stages of evacuation is more obvious. Because of the nature of the evacuating crowd, this paper chooses the SFPE mode in Pathfinder. SFPE mode can simulate the evacuation congestion scenario, which will be expanded in subsequent sections.

Several studies [30,35,36] have shown that selecting multiple crowd compositions and monitoring evacuation efficiency at each entrance/exit can better identify where evacuation needs to be improved in existing planes. However, this study is different from the above improvement methods. The purpose of this study is to adjust the location of the entrance and exit, observe the flow direction of people and the efficiency of the entrance and exit in the case of multi-type crowd combination, and verify the plane shape of the strip commercial street that is most conducive to evacuation. Of course, there are many ways to change and influence the flow of people, such as people's familiarity with the road and setting up signs [30,35]. To explore the influence of the change in entrance position on evacuation efficiency more purely, in this study, we assume that people are familiar with the entrance position, that there are obvious signs at the entrance, and that people will

choose the entrance according to the principle of proximity, which will be reiterated in the subsequent assumptions.

This study introduces a flowchart of the implemented simulation and analysis framework (shown in Figure 5) used to investigate the effectiveness of the evacuation of the strip underground commercial street. This study analyzes the population and the arrangement of entrances and exits to verify the viability of the chosen strip-shaped subway commercial street. According to the characteristics of the layout of the planned entrances and exits of strip commercial streets, this study changes the positions of the entrances and exits accordingly in the Pathfinder simulation software, along with the parameters of the standard number of people and personal designs. This study finds the layout of the entrances and exits with the fastest evacuation time after the simulations and analysis. Multiple scenarios are selected for feasibility demonstration and evacuation performance comparison. Within these scenarios, the floating of the total population and the adjustment of the number of elevator users are the two possibilities that best describe the evacuation nature of the subway. The examined layout mode of the entrance and exit of a strip subway commercial street will be decided by comparing and analyzing the simulation results.

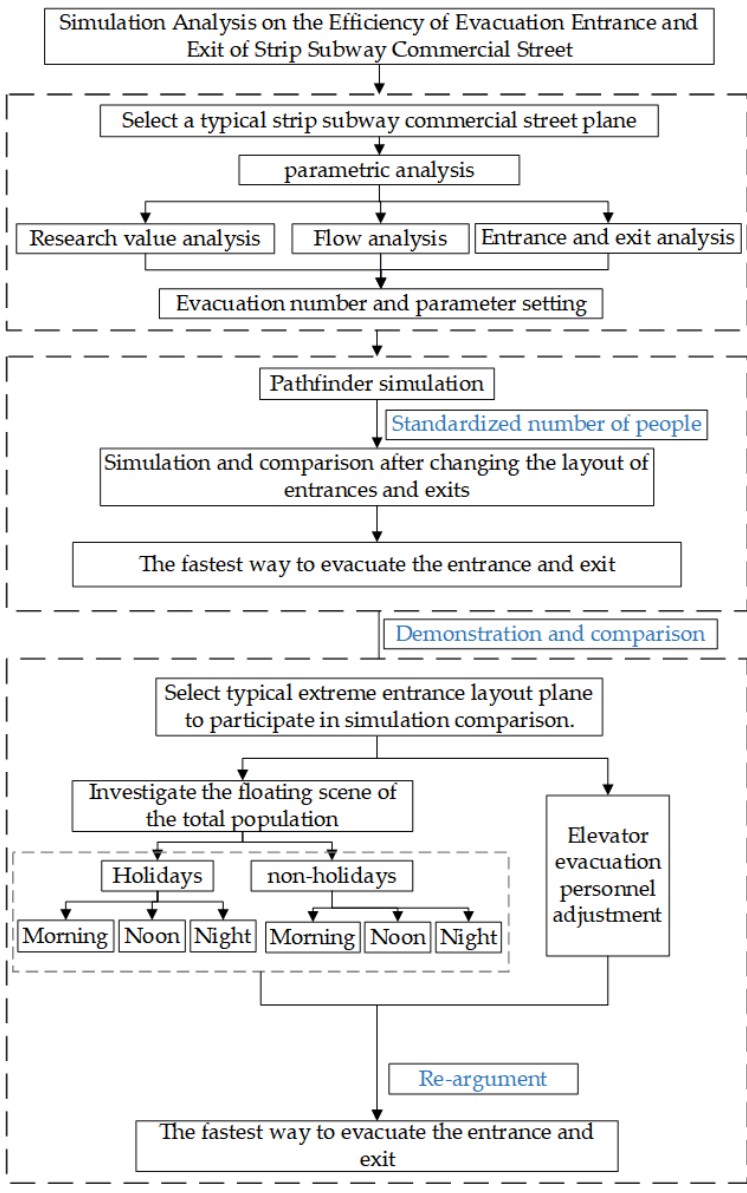

**Figure 5.** Flowchart of the simulation and analysis framework.

*3.2. Simulation Environment*

Pathfinder is a simple, easy-use, uncomplicated, agent-based evacuation simulation program. The software's parameters can be set by users, such as the body mass index, the rate at which objects are moving, and the number of individuals present in the scenario that needs to be reproduced [37]. The Pathfinder software has two operating modes: SFPE and Steering. SFPE uses some listed assumptions and a manual calculation mode described in the Engineering Guide to Human Behavior in Fire, whereas Steering is a guidance mode in which people who need to be evacuated use the guidance system to act and to interact with others. Although the door will limit the maximum flow of people and the speed is governed by the flow density, in SFPE mode, those being evacuated do not need to worry about preventing collisions with other individuals. To more accurately represent the evacuation features of a subway commercial street, this simulation has chosen to use the SFPE mode, which is more appropriate given a comparison between these two simulation models and the actual evacuation scenario of a metro commercial space.

The following basic assumptions are made based on the simulation environment:

1. Assuming that the confined individual is constantly acting rationally, there will not be any threats like trampling or slapstick; hence, the quickest route to the closest evacuation gate can be chosen.
2. Early in the evacuation process, when there are fewer people at the entry and exit, traffic will flow easily, and there will not be any congestion. The density of people at the entries and exits, however, progressively rose after a given amount of evacuation time, congestion started to occur, and the speed of transit slowed down or even temporarily stagnated.
3. The ignition position is situated in the lobby area of the subway platform, and the subway commercial street's escape and evacuation direction is from the lobby area of the subway platform to the exits of the subway commercial street.
4. The trapped people act immediately when the fire occurs, and we do not consider the time of judgment and start-up of the people in the simulation scenario.
5. Assuming that people are familiar with the location of the entrances and exits and that there are obvious signs at the entrances and exits, people will choose the entrance and exit according to the principle of proximity.

External influences such as evacuation instructions are not considered for the time being.

*3.3. People Design in the Simulation*

3.3.1. Standardized Number of People

According to the code conditions in the simulation environment, we need to estimate the number of evacuees for this commercial street. This study defines the calculated area of each floor of the business hall as $S_{bh}$ and the total floor area of each floor as $S_{gf}$. $S_{bh}$ can be estimated using Equation (1):

$$S_{bh} = S_{gf} \cdot K1 \cdot K2 \tag{1}$$

where K1 represents the scale correction coefficient and K2 represents the format correction coefficient.

The number of evacuees is defined as $N_e$, and the conversion coefficient of the number of evacuees on different floors of the commercial business hall is defined as E. The $N_e$ can be estimated using Equation (2):

$$N_e = S_{bh} \cdot E \tag{2}$$

After decomposing the formula and substituting the corresponding values, the data needed for the article are obtained.

Through field investigation, this study calculated that the construction area of the study area is 4587 m$^2$, the total evacuation width of the underground commercial street is 15.795 m, and the existing overall evacuation width is 50.2 m. In the study area, the underground building adopts the first and second fire resistance levels, in the definition of the relevant code, and the total commercial construction area belongs to the medium-sized

category. The K1 value is taken as 60%. This metro commercial street mainly contains shopping malls and restaurants, etc. The K2 coefficient is taken as 0.9, and the conversion coefficient of the ground floor is 0.85 people/m$^2$, according to the literature [2,26,30]. Therefore, the number of evacuees is set as 2106 people according to Equations (1) and (2).

### 3.3.2. Personnel Parameter Design

The analysis reveals that young and middle-aged women, young and middle-aged men, the elderly, and children make up the majority of those traveling through the subway commercial street in the Wuhan Business District. The number of participants in the simulation was set at 2106 and corresponded to the percentage of respondents to the survey. Young and middle-aged women made up half of the participants, making their total number approximately 1095. Young and middle-aged males made up the other half, at 42 percent; their total number was around 885. There are 21 kids and approximately 105 elderly adults. To produce a realistic model, the speed of young and middle-aged pedestrians is set at 1.5 m/s, the speed of elderly pedestrians is set at 0.9 m/s, and the speed of children is set at 0.66 m/s. The personnel parameters are listed in Table 2.

**Table 2.** Simulated evacuation parameters.

| Personnel Type | Proportion | Shoulder Measurement | Speed |
|---|---|---|---|
| Young and middle-aged women | 0.52 | 458 mm | 1.5 m/s |
| Young and middle-aged men | 0.42 | 486 mm | 1.5 m/s |
| Children | 0.01 | 336 mm | 0.66 m/s |
| Elderly people | 0.05 | 469 mm | 0.9 m/s |

### 3.4. Entrance Layout Simulation

The underground commercial street must adhere to two requirements to be permitted in the simulation environment. First, outdoor open spaces like sunken plazas should be considered when the entire building area of the neighborhood of stores is over 20,000 m$^2$, and the surrounding areas truly require local connectivity. Second, there should always be a minimum of two safety exits in each fire zone of a subterranean commercial street if several fire zones are close to one another. The subterranean commercial street in Wuhan Business District has the necessary number of fire zones, sinking squares, and safety exits, and the original number of entrances and exits can be kept for the simulation study.

To find the entrances of the strip metro commercial street with the fastest evacuation time, this study aims to relocate the evacuation points. In this regard, this study classifies the pattern of entrances as a symmetrical layout or a staggered layout depending on the shape of the underground commercial street. It can be divided into two sides or only one side depending on the orientation of the entrances and exits. Setting up open spaces like a sunken plaza is important when the local lower business street's total building area exceeds 2000 m$^2$. As a result, the current entrances and exits of the sinking square are marked, and they are divided into two categories: those that are in the middle and those that are leaning to one end of the entire strip-shaped underground commercial street.

The alternative floor plans shown in Figure 6 are based on the existing commercial street plan of the Wuhan Business District subway station, and the results are obtained after adjusting the entrance/exit positions with the original number of entrances and exits remaining unchanged, divided into two types of equal and unequal spacing according to the distance of entrances and exits and divided into two cases of the same direction and different direction according to the directions of entrances and exits. These scenarios will be further screened in the follow-up strategy comparison. In this study, the subway commercial street plan was drawn in CAD according to the typical entrance scheme divided, exported to a DWG format, then imported into Pathfinder software. Since the aim of the was to assess the evacuation situation of a subway commercial street, this study only extracted the components such as the floor, stairs, doors, entrances, and exits, then repaired the model and improved it according to the prompt information in the simulation software.

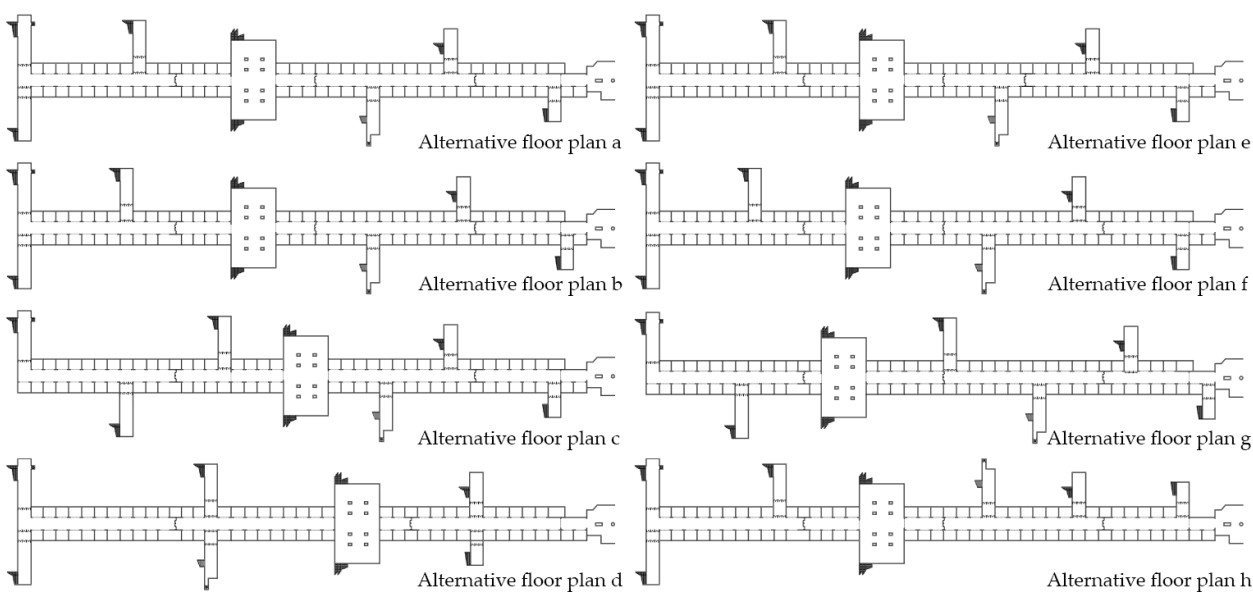

**Figure 6.** Entrance and exit scheme scenarios for the study area.

After constructing the basic model, this study distributed the rated number of people evenly according to different usage scenarios, set the characteristic parameters of people, and set up measuring tools at each entrance and fire zone so that the evacuation time of the subway commercial street under each situation in each typical entrance layout and the evacuation rate line chart of people passing through the entrance and fire zone could be obtained for subsequent research in Pathfinder.

### 3.5. Simulation Scenarios

#### 3.5.1. The Benchmark Evacuation Scenario

This study simulated the scenarios in Figure 6a–h with a total reference number of 2106 people and found the evacuation times of scenarios Figure 6a–h are 149.03 s, 143.78 s, 183.8 s, 186.8 s, 182.03 s, 151.03 s, 134 s, and 187.5 s, respectively.

This paper analyzed the evacuation time data and its corresponding plan shape, selected and retained the most extreme data, and traced it back to the corresponding entrance layout. The most extreme data were retained to analyze the difference in evacuation efficiency and obtain the most suitable layout of entrances and exits to improve evacuation efficiency. This study selected the entrance layout plan with the highest evacuation efficiency characteristics (e.g., the 95% evacuation time, the longest evacuation time, and the ratio of evacuation person/time). Finally, four representative plans were selected as the benchmark evacuation spaces according to the above selection criteria: (a), (d), (g), and (h) (shown in Figure 7). The figure include the corresponding symmetrical layout of all entrances and exits, the interspersed layout of all entrances and exits, and the layout of most unilateral entrances and exits.

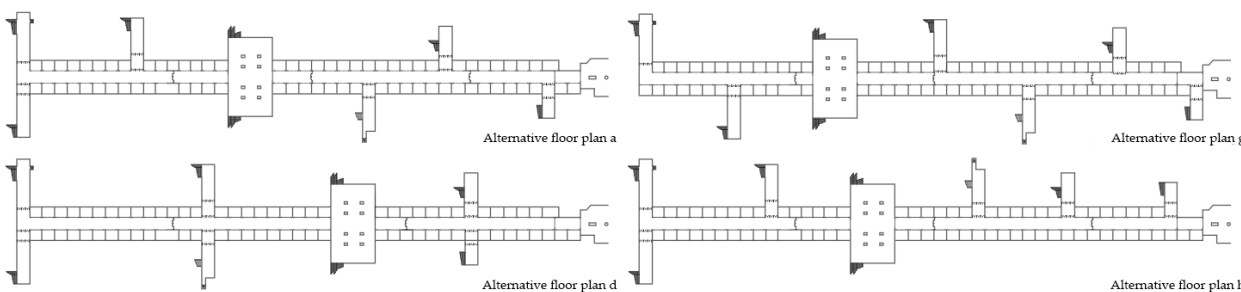

**Figure 7.** Alternative floor plans after the screening.

3.5.2. Floating Scenario of the Number of People to Be Evacuated

When the location of the metro commercial street in the Wuhan business district is known, there are 2106 evacuees. In the actual application situation, it is also required to consider the influence of the passenger flow on the metro commercial street as the intersection station of two metro lines in addition to the population computed from the area of the commercial street.

The metro lines 3 and 7 each have a fixed capacity of roughly 1440 passengers, with a maximum capacity of approximately 2000 people. Metro Line 3 uses a 6-section B-type car, while Metro Line 7 uses a 6-section A-type car, with a maximum capacity of 2528 people. Because the A-type car carries 500 more people than the B-type car, the personnel float in each train is between 2000 and 2500 people. The calculations for station visitors are represented in Equation (3):

$$Q = \lambda \Delta T(Q_1 + Q_2) \tag{3}$$

$$Q_1 = \max(Q_{up} - Q_{down}) \tag{4}$$

The relevant data details can be derived from Equations (4)–(6), where Q is, in individuals, the total number of people who must be evacuated in the case of a fire. During the forward or peak hour of the passenger flow control period, $Q_1$ is the single largest cross-sectional passenger flow, expressed in persons/h; $Q_{up}$ and $Q_{down}$ represent the number of passengers boarding and alighting the train; $Q_2$ represents the forward or peak hour incoming passenger flow on the up and down platforms during the passenger flow control period, and interchange stations also represent the flow of passengers from other lines switching onto the platform of the fire line, in persons/h; $\Delta T$ represents the peak hour interval in hours (h) during the forward or flow control period; and $\lambda$ represents the station's over-peak hour passenger flow factor, generally 1.1–1.4.

$$\Delta T = 1/N \tag{5}$$

$$N = Q_1/V\beta \tag{6}$$

where N stands for the number of train pairs that are now running, $\beta$ for the full train load factor, and V stands for the train's designed passenger capacity. Peak hours, such as in the morning and evening, will cause a significant variation in these statistics.

Part of the metro's passenger flow is diverted along the commercial street, which is one of the travel routes for the Wuhan Business District metro. Most of the establishments on this commercial strip are eateries, and due to the flood of people at breakfast, lunch, and supper, the population density rises considerably. One of Wuhan's most picturesque spots is the station, and it has ornamental value, boasting a sizable blue and purple "dome". Wangjiadun Park also draws tourists since it is a peaceful place. The two-hour morning, lunchtime, and evening passenger flows for both holiday and non-holiday situations are displayed in Table 3 after numerical calculations and field measurements of the metro passenger flows.

**Table 3.** The number of people passing through the subway commercial street in different periods.

| | The Number of Personnel/per. | | |
|---|---|---|---|
| Operational status\Time of day | 7:00–9:00 | 12:00–14:00 | 17:00~19:30 |
| During the holidays (4 August) | 2245 | 8650 | 13,205 |
| Non-holiday periods (16 August) | 6652 | 2184 | 9882 |

To more accurately simulate the usage scenario in life and to conduct a more thorough simulation of the entrance/exit layout of the metro commercial street, this study adds the business passenger flow in Table 3 to the simulation scenario based on the baseline number

of 2106 people so that the simulation approach can capture the typical business situation of the metro commercial street.

### 3.5.3. Adjust the Proportion of Elevator Personnel

When people with limited mobility use stairs to evacuate, it can have a blocking impact on other people, which can reduce the effectiveness of the evacuation to some extent. The gathering and waiting of individuals, as well as the intermittent evacuation, do occur, though, in elevators. If there are too many people using the elevators, it will clog up the entrances and exits, slowing down and lengthening the evacuation process.

There are two elevators in the Wuhan Business District subway station's current layout, one by entrance M and the other by entrance Q. Because it is at the end of the underground commercial strip and has a significant evacuation distance, the elevator at entry M is frequently neglected in simulation scenarios. More people will decide to depart immediately using the egress stairs at M. The elevator at the Q door receives far more frequent use than the elevator at the M entrance due to the complicated and congested traffic flow created by its location in the middle of the commercial street. This part will concentrate on the coordinated evacuation of elevators and staircases after a pre-screening of the model.

Following the observation of the layout of the four screened entrances and exits, the elevator at the Q entrance is designated as Q1. When there are 2106 evacuees and pedestrians are evacuated freely in all four layout options, there are 63 users of Q1 in the layout in (a), 81 users of Q1 in the layout in (d), 42 users of Q1 in the layout in (g), and 79 users of Q1 in the layout in (h). The waiting time and evacuation time of the final elevator have some bearing on the effectiveness of the evacuation process overall, according to a careful review of the outcomes of the simulated evacuation. Using the layout in (d) in Figure 7 as an example, the last person to take the stairs to evacuate vanished in 180 s, and it took 14.5 s for the elevator's six passengers to board it. The last person to evacuate using the stairs in the layout (g) of Figure 7 disappears at exit Q at 128.6 s, after which all four people in the lift are evacuated. The elevator evacuation in the layout (h) of Figure 7 setting evacuates five people after the last person who used it vanished at exit J at 150.2 s, which significantly reduces the efficiency of the evacuation process. This paper will set limits on how many people utilize the elevators and compute how efficiently each configuration may be evacuated while still maintaining the same level of elevator personnel use.

To study the rule of the cooperative evacuation of stairs and elevators, the simulation approach selects five situations with a different number of people using the elevator; that is, when the elevator is not in use, 6 percent, 10 percent, 15 percent, and 20 percent of the reference number (2106 people) evacuate using the elevator, and the remaining 80% evacuate using the stairs. What needs to be made crystal clear is that there are 126 elderly and children in the benchmark population, or 6% of the entire population. The number of individuals following each of the aforementioned percentages is immediately eliminated after the decimal point; rounding is not used.

## 4. Experimental Results and Analysis

The above simulation scenarios were run in the simulation approach to determine and repeatedly show the layout of the entrances and exits of the subway commercial street with the highest evacuation efficiency. The results of these simulations will show which layout position of the evacuation entrances and exits is the most beneficial for personnel evacuation.

### 4.1. Evacuation Results of the Benchmark Scenarios

When the number of passengers is used as the reference number in four typical evacuations after the screening, Figure 8 illustrates the link between the number of evacuees and the time required for each evacuation scenario. The prejudgment and response times

of the staff are not considered in this study. Every person who has to be evacuated and who is trapped will begin to act immediately once the fire starts.

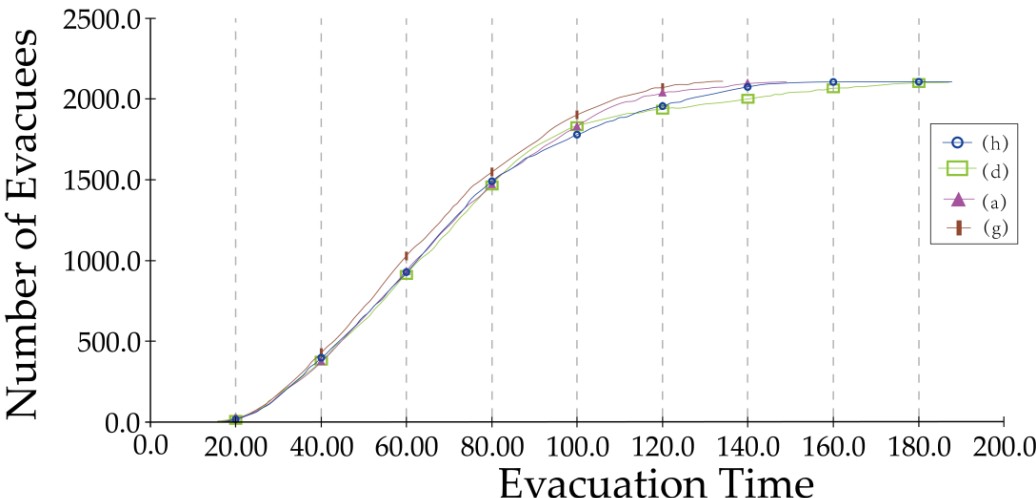

**Figure 8.** Relationship between evacuation time and number of evacuees in each layout.

According to the simulation approach's output, the four typical layouts all begin the first evacuation at a time of 15 s, which means that the distance between the exit and entrance is at its shortest in 15 s. The adjustment of the exit and entrance layout has no impact on the initial evacuation departure time.

When the time came to the 20th second of the simulation, there was no discernible change in the number of evacuees, and on the 40th second, the layout (g) arrangement started to rise, which was marginally different from the number of evacuees in other layouts. The number of evacuees at each time node was greater than it would have been in other layouts, as layout (g)'s evacuation effect grew in the subsequent period. However, there are not many differences, and they practically overlap in the first 80 s of the layouts (a), (d), and (h) in terms of the number of evacuees. The number of evacuees in (d) was higher than that in layouts (a) and (h) in the 1980s and 1990s because the slope of layout (d) remained constant in the 1980s, while in layouts (a) and (h), the slopes started to decline. However, there were inflection points at the 88th and 100th percentiles, which led to a decline in the slope and a progressive reduction in the number of evacuees. The efficiency of the evacuation fell off at 120 s.

There was a divergence point at 88 s between the evacuation efficiency of layouts (a) and (h) in the early stages. While layout (a) essentially maintained the original evacuation efficiency until a slow inflection point appeared around 107 s, the evacuation efficiency in layout (h) was sluggish after 88 s, leaving one person to be evacuated at 158 s until the last person at 187.5 s.

The evacuation efficiency of each plan changed somewhat in the first 40 s, 88 s, and 100 s, as seen by a comparison of the image information. By examining the evacuation circumstances of these three time nodes, the author seeks to ascertain the causes of the shift in evacuation efficiency. Figure 9 shows the numerical evacuation line for the 40 s, 88 s, and 100 s respectively.

The evacuation line makes it plain that layout (g) offers many advantages during evacuations, and there are certainly more evacuees than with alternative layout strategies. A glaring disparity can be noticed in the rate of the crowd gathering in the density map. The density is split into blue and red ends in Figure 10 below. Low personnel density is indicated by the color blue, which gradually changes to light blue, green, yellow, orange, red, and crimson as the personnel density steadily rises to a maximum of 3 people/m². In Figure 10a, it is obvious from the color of the density map that, compared with the first

40 s, only in layout (g) are the N exits in the layout mode in a fully utilized state, and the N exits in the other three layout modes are only used by scattered people.

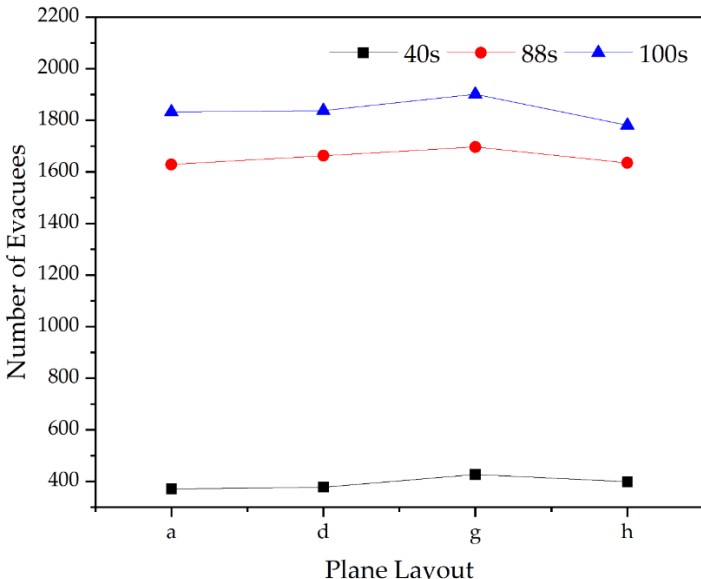

**Figure 9.** The number of evacuees corresponding to each plane layout at different time points.

Exploration reveals that this is the result of the simulation approach, which directs personnel to the closest entrance and exit, both of which are placed on either end of the layout. People in the simulation model will typically travel to the M exit after evaluating the walking distances of the entries and exits; hence, the N exit has not been utilized significantly. For the layout (g), each fire zone is divided into no more than 2000 m², and after the defined number of entrances and exits is calculated based on the crowd, the N exits are arranged in the middle position between the exit at the end of the subway commercial street and the entrances and exits of adjacent fire zones, and N exits are standardized in the original fire zones so that people in the middle can easily find a more convenient evacuation path.

Figure 10 shows the evacuation situation of layouts at the 40 s, 88 s, and 100 s marks, respectively. Layout (g)-2 shows that people are still using the N gate at the 88 s mark, which significantly increases the N gate's efficiency. Only one or two entrances and exits per layout remain at the 100 s mark, but evacuation still takes place through these. Combining the evacuation curve of Figure 8, Figure 9 shows that the number of evacuees in layouts (a), (d), and (h) is identical at 88 s, while the number in layout (d) is somewhat greater. The density chart of Figure 9 can be compared in real time to show that although the number of entries and exits in layout (d) is lower, the grouping of entrances and exits is a little too close together. Due to the crowding of people just at the frequently utilized entrances and exits, the evacuation efficiency is lowered and tends to be flat at this time. As a result of the overcrowding, vicious situations like trampling at the Q and J entrances and exits occur at the 120 s time point, when the number of evacuees is much smaller than that of layout (a) and (h).

This study compares how effectively each layout's entrances and exits are used in Figure 11. The efficiency of the use of the entrances and exits of the layout (g) as a whole is relatively concentrated, and the evacuation of each exit is completed in a similar time, without the situation that most of the entrances and exits have been running towards the end and a small number of entrances and exits are still widely used, except for Q1, which is a lift. The use curve of the N entrance of the layout (g) is shown in Figure 12, which is much higher than the use curve of the other layouts, with a shorter use time and higher evacuation efficiency. With the rational use of N entrances, the frequency of use of each entrance and exit reaches a relative concentration, making the overall evacuation time stay

within 140 s, while other layouts have low evacuation efficiency and even more directly exceed 180 s.

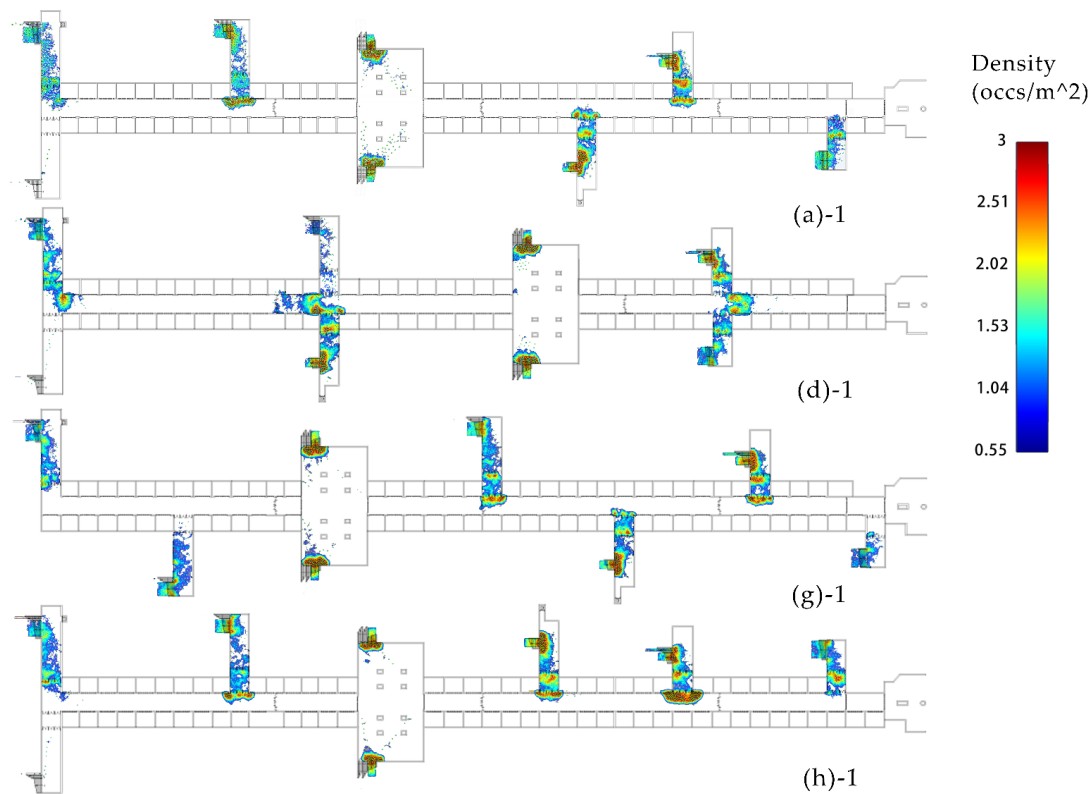

(**a**) The evacuation situations of layouts identified as (a)-1, (d)-1, (g)-1, and (h)-1 at the 40 s mark

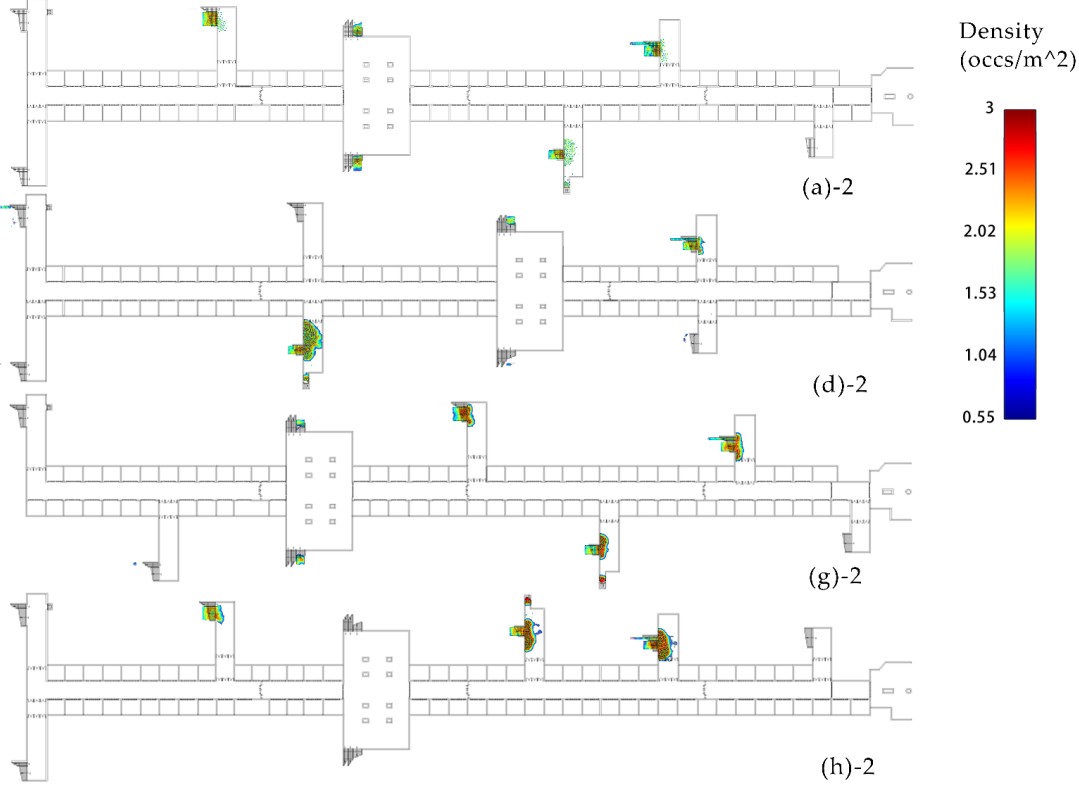

(**b**) The evacuation situations of layouts identified as (a)-2, (d)-2, (g)-2, and (h)-2 at the 88 s mark

**Figure 10.** *Cont.*

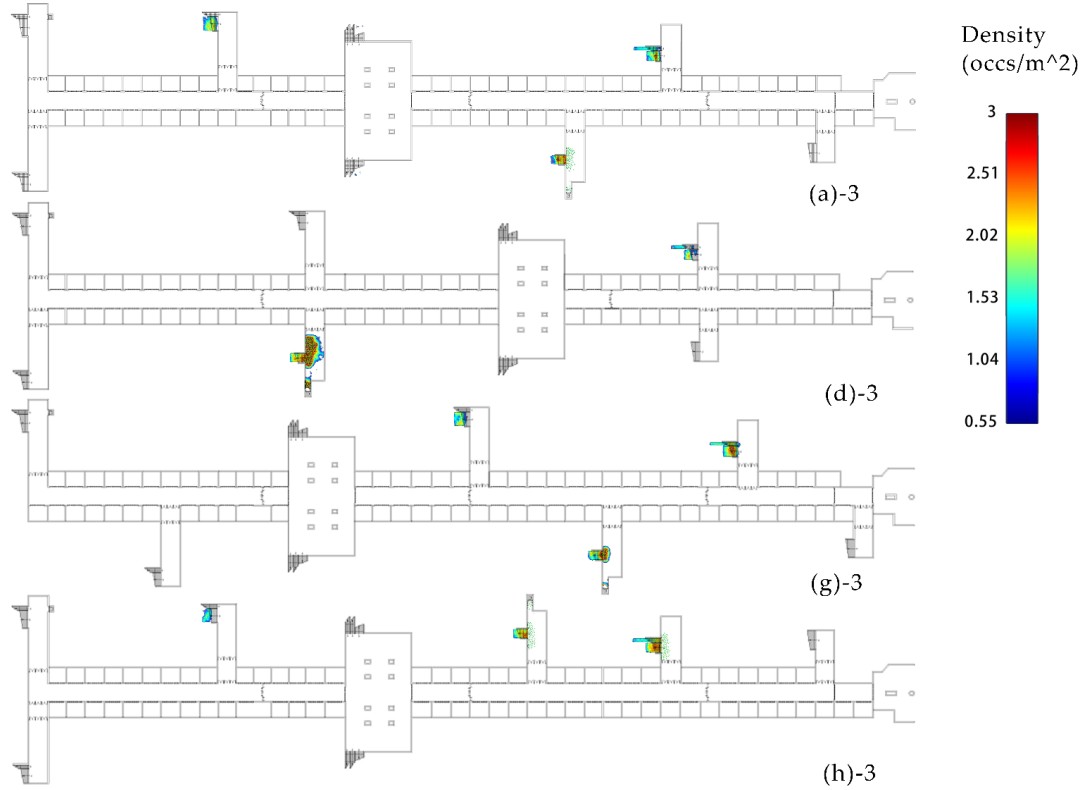

(**c**) The evacuation situations of layouts identified as (a)-3, (d)-3, (g)-3, and (h)-3 at the 100 s mark

**Figure 10.** The evacuation situation of layouts at the 40 s, 88 s, and 100 s marks, respectively.

Therefore, considering the evacuation time, the number of evacuees, and the number of evacuees for each period, this study can conclude that the equally spaced interspersed entrance/exit distribution represented by layout (g) is a more favorable entrance/exit layout for the evacuation of strip-like underground commercial streets, provided that the number of entrances and exits is the same and that the fire regulations are met.

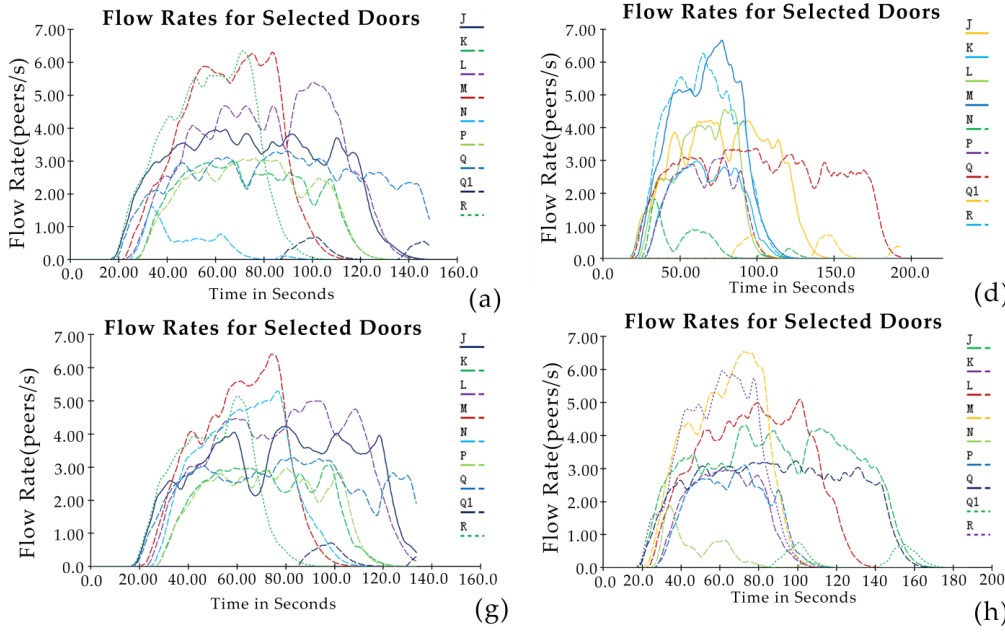

**Figure 11.** (a)-4, (d)-4, (g)-4, and (h)-4 correspond to the use of each entrance in (a), (d), (g), and (h) evacuation layouts, respectively.

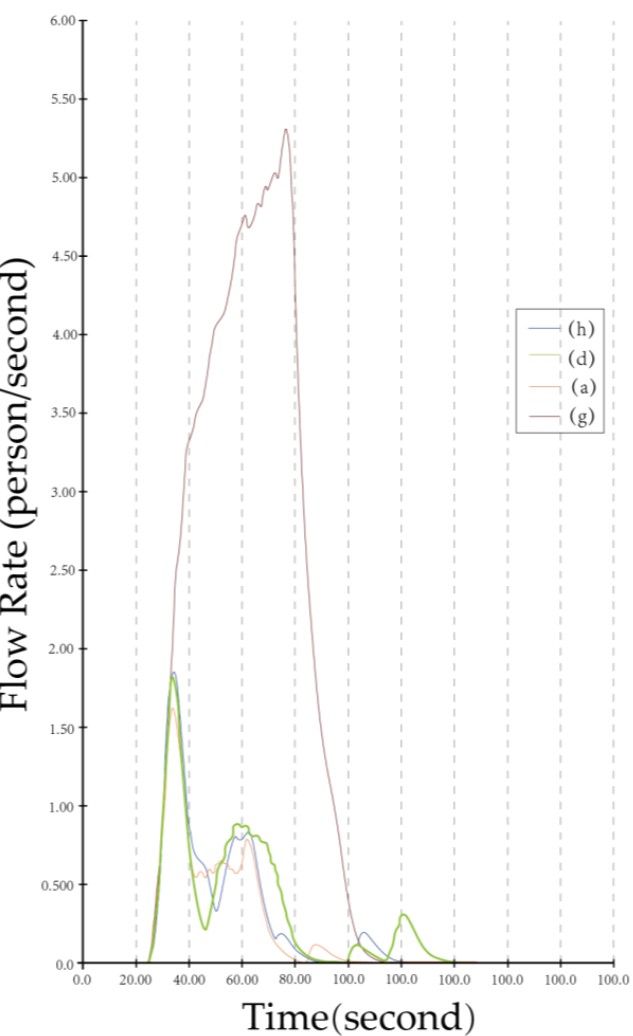

**Figure 12.** The n-inlet flow rate of each arrangement of inlets and outlets.

### 4.2. Evacuation Results after the Number of People Fluctuates

The simulation approach considers the floating scenario of the number of people to be evacuated in the morning and noon periods of workdays and weekends, respectively. Figure 13 shows a great difference in the evacuation time. It can be found that with the increase in the number of evacuees, the evacuation time is on the rise, but the growth in the efficiency of evacuation is not consistent. The figure shows that the time of the number of evacuees is relatively flat from 2106 to 2245, and it fluctuates up and down in the 200 s with a small fluctuation range. The time transformation of the number of people in the range of 6652–9882 did not fluctuate too much, and the evacuation time was controlled at about 400 s. When the number of people rises to 13,205, the evacuation times of layouts (a), (d), and (g) are relatively short, and in layout (h) when the evacuation time exceeds 1400 s, the evacuation efficiency slows down. Therefore, through simulation calculation, the evacuation time of the whole subway commercial street is controlled within 600 s. Since the commercial street is connected to the subway, there is no way to control the evacuation flow, so it is particularly important to choose the layout mode with the shortest evacuation time. Among the four arrangements, the evacuation time of layout (g) is mostly at the shortest end of the four evacuation durations, so it can be seen that in these subway commercial street plans, the evacuation efficiency of the equally spaced interspersed entrance distribution represented by layout (g) is also the highest.

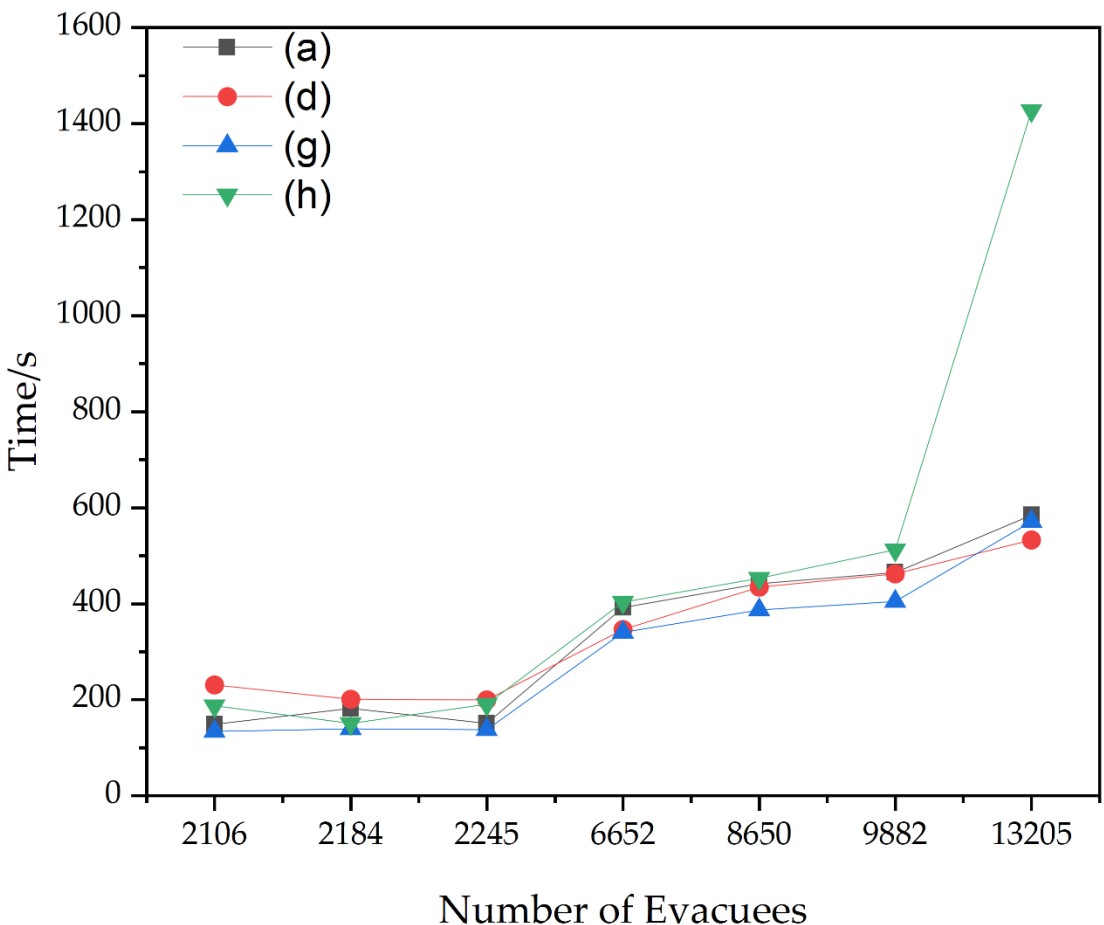

**Figure 13.** Changes in evacuation duration of different layouts for different numbers of people.

*4.3. Evacuation Results after Adjustment of Elevator Use Ratio*

After adjusting the number of elevator users in each entrance layout in four proportions based on the reference number of 2106, is the results are compared with the situation of not using elevators and using elevators freely, and the comparison results are sorted into a line chart in Figure 14. As can be seen from Figure 14, with the increase in the proportion of elevator users, the evacuation time shows a trend of slightly decreasing at first and then continuously increasing, and the more the elevator frequency is increased, the longer the evacuation time will be. No matter what the arrangement, the proportion of people using elevators is almost positively correlated with the time spent evacuating. The data show that the evacuation efficiencies of layout (d) and layout (h) are indeed improved when the elevator is not used, but in contrast, the evacuation time of layout (g) is the shortest. The evacuation efficiency of layout (g) is the highest in all simulation scenarios under the condition of the free use of elevators, and it only takes 134 s to evacuate the same number of people.

Because the commonly used elevators are concentrated near the Q1 exit, there is a certain distance between the Q1 and Q exit. After carefully observing the running trajectories of various evacuation models, the author finds that after the number of users is specified, people will aggregate to wait for the running elevator without choosing the relatively empty and faster stairs, which greatly delays the evacuation time. When it is stipulated as free use, people can reasonably choose whether to take the stairs or the elevator according to the principle of nearby selection, thus reducing the blocking behavior to some extent. When in free use, the number of elevator users in layout (g) is 42, accounting for about 2% of the total number. The premise of this situation is that it does not change the original combination distance of the elevators and stairs and the evacuation distance

from entrances and exits. Therefore, in this case, only Q1 has a use efficiency, and the use efficiency is extremely low. If we want to further improve the efficiency of elevator use, we can consider increasing the number of elevators and making them closer to the entrances and exits.

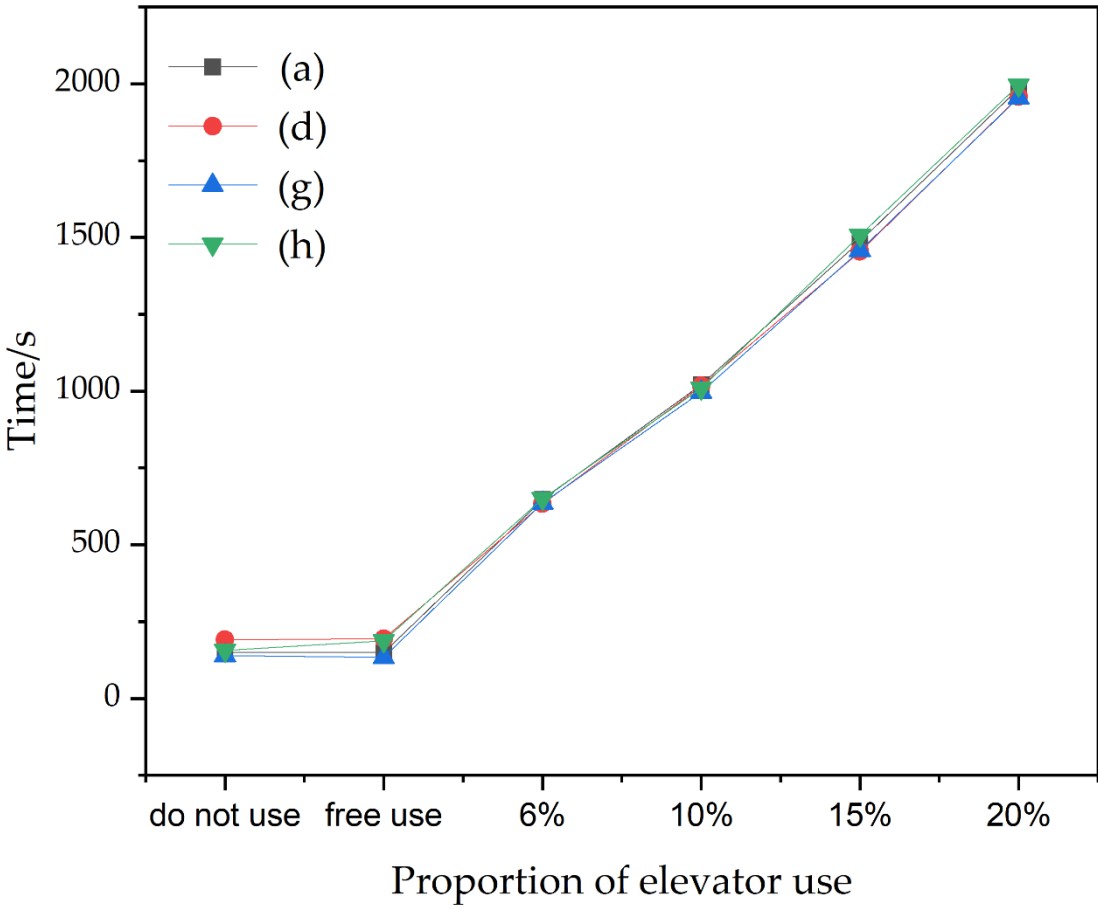

**Figure 14.** The elevator use in each arrangement.

As this study focuses on the influence of the location of entrances and exits of strip metro commercial streets on the evacuation of people, only the grouping and adjustment of the corresponding entrances and exits are considered, without focusing on the detailed arrangement of the internal facilities of the entrances and exits. This indicates that layout (g) has the superior layout relationship between entrances and exits, with equal intervals after meeting the fire code, which is more conducive to improving evacuation efficiency.

**5. Discussion**

By analyzing and comparing the above representative experimental results, we can know that changing the entrance layout will affect the evacuation efficiency of a strip-shaped subway commercial street. This is consistent with the theory of many studies [22,24,38]. The research results show that in the face of various situations, the evacuation efficiency of equally spaced, interspersed entrances and exits is faster than other layout schemes. This is contrary to some research conclusions. Some scholars [22,24] think that symmetrical double exits are more conducive to evacuation, and the change in the layout of entrances and exits they mentioned has brought innovation to this study. However, their research scope is very different from that of this study. This study has concluded from many situational experiments that the most effective layout of evacuation exits in strip-shaped subway commercial streets is the equally spaced interspersed layout, without changing the number or widths of entrances and exits. It is worth noting that this

study focuses on the layout of the entrance and exit of the subway commercial street itself, rethinking the location of the entrance and exit from the perspective of evacuation and design, rather than optimizing and guiding the evacuating people based on the existing plane shape [23]. There are some drawbacks to this field research, and all types of users are not considered in the personnel selection. In the follow-up research, priority will be given to how the plane's entrances and exits are related to spatial recognition, thus bringing greater evacuation convenience to different types of people.

## 6. Conclusions

Due to its traffic attributes, the subway commercial street cannot limit the use of passenger flow, and the evacuation efficiency is of great importance in the layout. The requirements for evacuation safety of metro commercial streets are increasing, and the reasonable arrangement of entrances and exits can effectively enhance the evacuation capacity of metro commercial streets. This study introduces a simulation and analysis framework to analyze the evacuation efficiency of a strip-shaped subway commercial street with the same number of entrances and exits and different entrance positions. This study takes the subway station in Wuhan Business District as the study area and considers the corresponding composition of the evacuation flow. The study simulates the study area's evacuation capacity, compares and analyzes the influence of different entrance and exit arrangements on evacuation efficiency, and demonstrates it from the perspectives of evacuation comparisons of different reference numbers, the floating number of people to be evacuated, and differing uses of elevators. The main conclusions are as follows:

1.  When the number of entrances and exits is the same, the layout of equally spaced entrances and exits represented by layout (g) is superior to other simulated entrances and exits in terms of simulation speed and the utilization efficiency of each entrance and exit, demonstrated by the evacuation density map. The layout (g), with entrances and exits interspersed at equal intervals, has the highest evacuation efficiency, which is 2–12% shorter than that of other layouts. The equally spaced interspersed entrances and exits have the characteristics of a uniform distribution, a short evacuation distance, and easy selection, which can make the utilization ratio of each entrance and exit more balanced.

2.  In the aspect of exits, the evacuation speed of exit N in layout (g) is increased by 15%, and the evacuation time is shortened by 10%. The overall evacuation time is shortened by 15 s, which is 96.8 s shorter than that of the symmetrical entrance layout, which has the longest time. Therefore, when the strip-shaped subway commercial street area is fixed and the number and widths of the entrances and exits are not changed, the change in entrance position will change the evacuation efficiency.

3.  When the number of elevators is small, free evacuation can enable people to choose a suitable route quickly. For the six selected elevator users, the evacuation time is positively correlated with the number of elevator users; that is, the more people use the elevator, the longer the evacuation time will be. Meanwhile, the free evacuation does not limit the number of elevator users, thus reducing the elevator waiting time. Among the six elevator users, layout (g)'s free evacuation mode is still the fastest evacuation layout form, which only takes 134 s.

However, in the process of evacuation, the default evacuation personnel are in a rational state, and the impacts of stampeding and crowding reactions on evacuation efficiency in a panic state are not considered, which will be further explored in future research. The specific element focused on here is only the evacuation entrances. The effect of other components will be discussed in our future work, such as the clues of spatial orientation. Also, more variants of the social diversity of subway users will be tested in the future, which will be helpful to evaluate the real performance of the strip-shaped subway commercial street. This study will put forward corresponding improvement measures for the design layout of the entrances and exits of strip-shaped subway commercial streets, which will provide data and theoretical reference for the layout of subway commercial streets and provide

effective technical support for improving the evacuation efficiency of strip-shaped subway commercial streets.

**Author Contributions:** Conceptualization, M.C. and H.X.; methodology, M.C. and H.X.; software, M.C.; investigation, M.C.; writing—original draft preparation, M.C.; writing—review and editing, H.X. and M.C.; project administration, H.X.; supervision, H.X.; funding acquisition, H.X. All authors have read and agreed to the published version of the manuscript.

**Funding:** This work was funded in part by the National Natural Science Foundation of China, grant number 41771473; in part by the Hubei Construction Science and Technology Project, grant number [2022]2198-123; and in part by the Cooperative education project of the Ministry of Education, grant number 22067124261715.

**Data Availability Statement:** Not applicable.

**Conflicts of Interest:** The authors declare no conflict of interest.

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
