# Peer review of "Simulation Analysis of the Efficiency of Evacuation Entrances of Strip-like Metro Commercial Streets: A Case Study of Wuhan Business District Metro Station"

_buildings, doi:10.3390/buildings13071826_

Round 1
Reviewer 1 Report
Generally, the outcomes presented in this manuscript are interesting and the study was properly executed. There are major shortcomings related to the communication of the results, which makes this manuscript not representative of a scientific text.
First and foremost, it is very worrying that all references come up in the first three paragraphs of the text. It makes an impression as if this text was a report, which was prepended with three paragraphs of an introduction and sent for peer review in a scientific journal. There are also serious flaws in the illustrations submitted for review.
The biggest concern related to the illustrations is image rights. Figure 1 uses an online map as a source. Basemap source should be indicated and followed by a statement that authors have the right to reuse the work in their publication. Alternatively, open-access base map data should be used (e.g. OpenStreetMap) or the authors should prepare their own illustration to be used in the manuscript. In this regard, Figure 2 is even more worrying, since it looks like a screenshot or picture taken from an illustration done for the party that owns and operates the subway infrastructure. It is advised to indicate the source of this illustration, securing rights to publish this image in advance. Alternatively, authors can prepare their own, simplified version of this 3D diagram, based on the 2D information they have available. From the purely graphic perspective, the off-white background should be removed in Figure 2 and the image should be submitted with transparent background (.png) with the required resolution.
It is advised to restructure the manuscript to contain the Methods section. Some sub-sections of section 3 could be assigned to the Methods section. Most importantly, the method description requires relevant references to be cited. Methods introduced in this study should be discussed in relation to the state of the art to elaborate how the methods introduced go beyond the current status. It should be stated what are the limitations of existing methods that were addressed by the method introduced in this study.
Figure 5 contains a simulation and analysis flow chart. Conventionally, flow charts do not contain elements which are not a part of a computational procedure. For this reason, discussion and conclusion should be removed. The smallest font size used is not readable. Please increase the size of the texts "Standarized number of people", and other texts in the same size.
In line 184, the numbered list does not use the style from the MDPI template. For this reason, it "sticks" to the paragraph above and below. Please use the proper formatting from the MDPI template.
In lines 206-219, formatting of the equations, variable descriptions and the paragraph in between is wrong, please consult the MDPI template for an example.
Tables 2 and 3, their captions and the paragraphs after are using the wrong formatting, please consult the MDPI template for an example.
The term "Optimization Plane" needs to be introduced and explained properly before it is used in Figure 6. Consider a different term that comes up more often in literature. Are those "optimization planes" "different scenarios", "design alternatives", "alternative floor plans", "alternative entrance locations"?
Regarding fonts in illustrations, consider using the freely available Palatino Linotype, which is the font used in the MDPI Word template. This could make your illustrations well-aligned graphically with the overall layout. Alternatively, please decide on a single, well-known font family (e.g. Arial, Calibri, Roboto) that will be used across all illustrations.
Figure 6 - consider removing frames around each floor plan image. The text "optimization plane ..." uses a different font which is hardly readable.
Figure 7 - caption does not communicate the contents of the figure since it consists of only two words.
Figure 8 - please match the font used and use the required MDPI styles for the figure caption and following paragraph. Check the capitalization of the words in the graph axes.
Figure 9 - please match the font used and use the required MDPI styles for the figure caption and following paragraph. Check the capitalization of the words in the graph axes. Images look like they have been distorted due to uneven scaling in the x and y directions. Please avoid it at any cost. It would be advisable to make sure that the whole figure fits on a single page. Consider placing the three graphs in a single illustration and add the texts (a)40s, etc. in the image editing program.
Figure 9 - consider removing frames around each floor plan image. It would be advisable to make sure that the whole figure fits on a single page. Colour scales are unreadable since they are too small. If the range is the same on each illustration, the colour bar should be added only once, which gives it more place on the layout and increases readability. Consider placing app floor plans in a single illustration and add the texts "(b)The evacuation situations of layout identified as (a)-2, (d)-2, (g)-2, and (h)-2 at the 88s.", etc. in the image editing program.
Figures 13 and 14 - should be provided without the frame. Please match the font used in all illustrations. Graphic consistency of the graphs is advised, which refers to the graphic representation of the axes and colours used in the line plots.
Lastly, a discussion of the presented results is missing. It is advised to discuss the authors' own findings in relation to the state-of-the-art contributions in the field. The discussion should contain references to the works published in this domain of study. Discussion should indicate: (1) what shortcomings of the existing approaches were addressed in this research. (2) which of those shortcomings were successfully overcome by the authors? (3) if there are any, list the shortcomings that were not successfully resolved by the authors and (4) indicate the future outlook (new challenges that are not addressed currently) for the research in this field. Style provided by the MDPI template for numbered lists should be used when listing the main conclusions.
Reviewer 2 Report
The article discusses a very important problem of evacuation from public places, here on the example of the Wuhan Business District subway station. This is a research problem as well as an implementation problem. The presented study corresponds to the profile of the journal. In the introduction, the authors of the study present the research background. A clearly defined goal is missing, please complete. Methods and materials well described, test results clearly presented. There is no discussion, please complete. The conclusions are not surprising, but supported by research; can be described in more detail.
Reviewer 3 Report
Dear Authors,
The presented text is a well-constructed scientific article addressing the important and current issues of evacuation of transport hubs in a specific localisation. Overall, I give a good assessment of the study and its report. However, I have a few comments that I think need to be added.
1. Spatial orientation is a complex process based on a number of components. One of them is the readability of space (including clues, affordances and anti-affordances encoded in it). Nevertheless, other elements that help intuitively make decisions related to mobility are also important, for example, an information and signage system, multi-sensory warning and evacuation systems, instrumentation and services supporting evacuation in emergency situations. In a word, I lack information about the fact that a specific element is being tested among many other components affecting the efficiency of evacuation.
2. I find a similar gap in your treatment of the social diversity of subway users. There is a difference whether during the evacuation we will be dealing with a heterogeneous, mixed crowd, or whether at a given moment it will be a large group of children or, for example, people with limited mobility. It is worth emphasizing that the situation that you take as a model for the simulation does not exhaust the various variants.
3. The structure of the article lacks a discussion point, which in my opinion should be included in a text of such a practical nature. Currently, I only find the conclusions section and in them something like a summary of the entire text. This part should be added, indicating what values ​​result from your research, what and in what situations the conclusions can be used, what are the limitations of the implementation possibilities. In addition, how do your observations differ from what other researchers have done, and which conclusions are current.
Kind regards
Round 2
Reviewer 1 Report
Dear Authors,
I would like to thank you for investing substantial time and effort in reviewing your manuscript. I appreciate your precise answers to my comments and I would like to congratulate you for reaching a level of a good scientific publication. Your article has reached a level at which it can be published in Buildings very soon. I would like to offer you my last few comments, which are more of editorial nature. I am convinced that many of the points raised can be solved in collaboration with MDPI editors.
Please discuss a layout strategy for page 5, in particular, the text of the caption (3. Simulation and Methods) should be placed on page 6. Figure 5 is substantially improved, I would consider moving the two upper, blue texts to the right, so that they don't intersect with the black lines. Equation formatting is correct in the file with your answer, but it is wrong in the updated PDF provided by MDPI. Please make sure that the formatting is correct in the final version of the manuscript. In the caption of Figure 7 please consider updating the text to: "Alternative floor plans (...)". Lastly, on pages 15, 16 and 17, in Figure 10 please consider adding space after (a), (b) and (c).
